# Visualizing Sacbrood Virus of Honey Bees via Transformation and Coupling with Enhanced Green Fluorescent Protein

**DOI:** 10.3390/v12020224

**Published:** 2020-02-18

**Authors:** Lang Jin, Shahid Mehmood, Giikailang Zhang, Yuwei Song, Songkun Su, Shaokang Huang, Heliang Huang, Yakun Zhang, Haiyang Geng, Wei-Fone Huang

**Affiliations:** 1College of Animal Science (College of Bee Science), Fujian Agriculture and Forestry University, Fuzhou 350002, China; 1170744011@fafu.edu.cn (L.J.); shahid@xtbg.ac.cn (S.M.); Zhang598214762@163.com (G.Z.); Song1152731522@163.com (Y.S.); susongkun@zju.edu.cn (S.S.); skhuang@fafu.edu.cn (S.H.); xiaoliangfujian@163.com (H.H.); yakun.zhyk@gmail.com (Y.Z.); haiyang_fafu@163.com (H.G.); 2CAS Key Laboratory of Tropical Forest Ecology, Xishuangbanna Tropical Botanic Garden, Chinese Academy of Sciences, Kunming 650000, China; 3College of Life Sciences, University of Chinese Academy of Sciences, Beijing 100049, China

**Keywords:** Chinese Sacbrood, iflavirus, cloned virus, expression tag, 3′-UTR

## Abstract

Sacbrood virus (SBV) of honey bees is a picornavirus in the genus *Iflavirus*. Given its relatively small and simple genome structure, single positive-strand RNA with only one ORF, cloning the full genomic sequence is not difficult. However, adding nonsynonymous mutations to the bee iflavirus clone is difficult because of the lack of information about the viral protein processes. Furthermore, the addition of a reporter gene to the clones has never been accomplished. In preliminary trials, we found that the site between 3′ untranslated region (UTR) and poly(A) can retain added sequences. We added enhanced green fluorescent protein (EGFP) expression at this site, creating a SBV clone with an expression tag that does not affect virus genes. An intergenic region internal ribosome entry site (IRES) from Black queen cell virus (BQCV) was inserted to initiate EGFP expression. The SBV-IRES-EGFP clone successfully infected *Apis cerana* and *Apis mellifera*, and in *A. cerana* larvae, it was isolated and passaged using oral inoculation. The inoculated larvae had higher mortality and the dead larvae showed sacbrood symptoms. The added IRES-EGFP remained in the clone through multiple passages and expressed the expected EGFP in all infected bees. We demonstrated the ability to add gene sequences in the site between 3′-UTR and poly(A) in SBV and the potential to do so in other bee iflaviruses; however, further investigations of the mechanisms are needed. A clone with a desired protein expression reporter will be a valuable tool in bee virus studies.

## 1. Introduction

Sacbrood virus (SBV) is a common virus belonging to the genus *Iflavirus* and causes failure to pupate and subsequent death in *Apis mellifera* [1,2,3]. It was the first identified iflavirus in European honey bees and a strain has evolved that also infects Asian honey bees, *Apis cerana* [4,5]. The *A. cerana* strain (Ac-SBV) shares more than 97% similarity with *A. mellifera* SBV isolates and causes more losses in *A. cerana* colonies than SBV in *A. mellifera* [4]. *A. cerana* is naturally resistant to *Varroa* mites and less prone to some common virus diseases found in *A. mellifera*; however, *A. cerana* apiaries in China were reporting serious losses caused by SBV [6]. Accidental introductions or natural spread of Ac-SBV into a new habitat often correlates with large-scale declines of both domestic and wild *A. cerana* populations (4). Moreover, the disease has spread to most natural habitats of *A. cerana* [5].

Multiple virus infections are common in individual bees [1,7]. Iflaviruses that may occur in mixed infections include SBV, Deformed wing virus (DWV), and *Varroa destructor* virus-1 (VDV-1; also called DWV-B). Other similar RNA viruses, e.g., dicistroviruses, include Black queen cell virus (BQCV), Acute Bee paralysis virus (ABPV), and Israel acute paralysis virus (IAPV). These viruses have similar size and sedimentary factors [8]. Viruses in mixed infections are difficult to separate using ultra-centrifuge methods. Without an uncontaminated isolate, it is difficult to study honey bee virus pathogenesis and the interactions among viruses, vectors, and hosts [9]. An infectious cloned virus with a distinguishable mutation may provide a practical solution to this problem [9,10,11,12].

Creating a clone of honey bee iflaviruses with desired nonsynonymous mutations is not straightforward. The reverse genetic data for honey bee viruses are still scarce compared to that of human picornaviruses or other animal picornaviruses causing economic loss. Sequence similarities are not sufficient to predict functional sites in the compact genomes of picornaviruses that have only one open reading frame (ORF) and require complicated cleavage processes after translation [13]. Adding sequences or nonsynonymous mutations into the ORF could easily disrupt the cleavage processes, resulting in inviable clones. Previous bee virus clones added synonymous mutations that needed identifications using RT-PCR [10,11,12]. A virus with such mutations is not easy to reveal in a living host or histological exam. Adding a florescent protein expression can solve the problem, but it has never been done in cloned bee viruses. Moreover, SBV does not have the feature sequence, 2A-mediated ribosome skipping, that can be utilized to fuse a green florescent protein (GFP) tag in a picornavirus [14]. In preliminary trials to attempt to create mutated clones, we found that our clones became inviable after introducing an additional sequence to the ORF. The only viable mutated clone was one with an added restriction-enzyme site at the end of the cloned SBV genome, between the 3′ untranslated region (UTR) and poly(A) tail. This result suggested that the site maintained an added sequence after cloning. Although picornavirus 3′-UTR is involved in viral genome replications, the sequences are varied in different viruses [15]. The secondary structures of 3′-UTR may be functional in replication [16], but no studies have suggested that the 3′-UTR terminal sequences are unchangeable in all picornaviruses [17]. We thus decided to exploit this feature of SBV 3′-UTR to create a clone with a reporter gene that does not mutate sequences in the single ORF.

We added a BQCV intergenic region (IGR) internal ribosome entry site (IRES) to mediate the translation of EGFP (enhanced green fluorescent protein) gene, based on the assumption that an IRES is needed to express a gene in the site. BQCV is a ubiquitous dicistrovirus that has two ORFs separated by an IGR-IRES and also infects honey bee larvae. The created clone has a bicistronic arrangement of two ORFs: SBV genome [ORF1]-BQCV IGR-IRES-EGFP [ORF2]-poly(A) (Figure 1). The cloned viral RNA successfully infected larvae by injection, and the reproduced SBV clones were isolated and passaged in *A. cerana* larvae by oral transmission, the typical SBV transmission pathway. Orally inoculated larvae had high mortality, and sacbrood-like symptoms were noted, fulfilling Koch’s postulates. To confirm repeatability and reliability of the clone, we used *A. mellifera* pupae because SBV-free *A. cerana* bees were not available later during the study, and the results were identical. EGFP fluorescence was observed in all clone-infected bees under a fluorescence microscope or by means of Western blots. Thus, our results indicate that the methods used in this study are repeatable and that the clone is sufficiently stable and infectious for further applications.

## 2. Materials and Methods

### 2.1. Virus RNA Extraction and cDNA Synthesis

*Apis cerana* colonies in Fuqing, Fujian, China, were screened for sacbrood virus and larvae with obvious signs of SBV infection, brownish color with liquified tissues, were collected and stored in RNAwait (Solarbio, Beijing, China). The samples were perfused at 4 °C for more than 24 h and then kept frozen at −20 °C. The larval samples were individually homogenized and RNA was extracted using Transzol (TransGen, Beijing, China), following the manufacturer’s instructions. The RNA samples were quantified using Nanodrop (Thermo-Fisher, Delaware, DE, USA). Two micrograms of RNA from the samples were used for reverse transcription with TranScript (TransGen, Beijing, China), one-step gDNA removal, and cDNA synthesis supermix, with oligo dT and random primers.

### 2.2. cDNA Template of SBV Genome

The SBV genome was separated into four fragments and PCR-amplified using primers (Appendix A). We used the Fuzhou AcSBV isolate (GenBank: KM495267) with a long 5′-UTR [18] as the reference genome for primer designs. Phanta DNA Polymerase (Vazyme Biotech, Nanjing, China) with proofreading activity was used, and the four genome fragments were cloned into pEasy T5 blunt cloning vector (TransGen, Beijing, China). We used the cloned fragments as the DNA templates to produce fragments with specific overlapping regions and modifications using anchored PCR for assembly. The 5′ end of the SBV genome (not suitable for designing primers) and oligo dT (25 bp; for producing the polyA tail of the RNAs transcribed using T7 in vitro transcriptase) was added using anchored PCR. Relative locations of the primers on the cDNA templates are shown in Figure 1. All resulting PCR fragments were electrophoresed and isolated from gels using EZDNA^®^ Gel Extraction Kit (Omega Bio-tek, Inc., Norcross, GA, USA).

We added an IRES and EGFP at the end of the cloned genome (Figure 1). An intergenic IRES sequence was cloned from BQCV, obtained from *A. mellifera* samples collected in the University of Illinois at Urbana-Champaign apiary. The EGFP sequence was obtained from the EGFP expression vector (GenBank: LC337090).

The Gibson assembly method was used to assemble the cloned fragments of the virus genome [11]. Blunt E1 Vector (TransGen, Beijing, China) with T7 promoter and terminator sequences was selected as the vector in *E. coli* (HB101). A pEASY^®^-Uni Seamless Cloning and Assembly Kit (TransGen, Beijing, China) was used to perform the Gibson assembly. The amount of each DNA fragment was determined according to fragment sizes, approximately 100 ng per 1000 bp. The mixture was incubated at 50 °C for 50 min, and the product was transfected into HB101 competent cells using the standard heat shock protocol. One bacterial colony was propagated in liquid LB with ampicillin, and the plasmid was isolated using a QIAprep Spin Miniprep kit (Qiagen, Hilden, Germany). The sequence of the plasmid was confirmed using Sanger sequencing (ABI 3730XL; performed by Biosan Biotech, Shanghai, China). The sequence has been submitted to GenBank (MN528599). A T7 High Yield Transcription Kit (Vazyme Biotech, Nanjing, China) was used for in vitro transcription. The viral RNA fragments produced (Appendix A) were precipitated using 2.5× volumes of isopropanol and then stored in 75% ethanol at −80 °C until use.

### 2.3. Inoculation Trials Using Apis Cerana Larvae

We selected three *A. cerana* colonies in late 2017 that had no detectable viruses, including SBV, DWV, BQCV, and IAPV, in larval and pupal stages as the sources of test larvae, using the screening method we previously reported [19]. One colony was used for viral RNA injection (P0) and the first oral inoculation passage (P1) was conducted in 2017. The two additional colonies were used in late 2017 to early 2018 for the following two oral inoculation passages, P2 and P3. We chose 3–4 day post-hatch larvae from selected brood frames that were held in a heated room (34 °C). Larvae were reared in 24-well plates placed in a sealed container with saturated NaCl water in a growth chamber at 34 °C. The larvae were fed and monitored following the protocol of Wang et al. (2009) [20]. Approximately 5-day post-hatch larva (after one day in a 24-well plate) were selected for RNA injection because of ease of manipulation. We injected 700 ng viral RNA in a 2 µL volume water suspension through a tracheal spiracle using a micro-injector (PLI-100A, Warner Hamden, CT, USA.) with a micromanipulator (WPI, Frankfort, Germany). Larvae in the control group were injected with nuclease-free water. After injection, the larvae were transferred into a 24-well plate and harvested 3 days post-inoculation before entering the prepupal stage. We homogenized the injected larvae individually in 200 µL PBS; 50 µL homogenate was used for RT-PCR and 20 µL for fluorescent exams. The homogenate of one selected larva was centrifuged at 10,000× *g* for 5 min to crudely isolate the cloned virus. The supernatant was used directly as oral inoculum for the next passage (P1).

Horizontal transmission by ingestion is the primary transmission route for SBV (1); therefore, oral infection is an essential indicator of our cloned SBV meeting Koch’s postulates. Because there is no clear dosage for SBV to reach 100% infection rate (ID_100_) in larvae, and we planned to enrich the cloned virus through the passages as well, we used 1 µL of the crude virus isolation (approximately 1/200 volume of the diseased larva) as the inoculum. Four-day-old larvae (*n* = 48) were randomly selected for inoculation in each of the oral passages (P1 and P2). A group inoculation method was used because of the small volumes and the stickiness of the diet. Each group consisted of 5–6 larvae, and each larva received approximately 4 µL diet, consisting of either 1 µL crude virus isolation or PBS solution. One heavily diseased larva was selected from the previous passage and used to isolate the cloned virus for each of the next two passages. Inoculations were terminated after the third passage because we were not able to obtain SBV-free *A. cerana* colonies after mid-2018.

### 2.4. Inoculation Trial Using A. mellifera Pupae

*Apis mellifera* pupae were used as an alternative host in 2019 to test repeatability and obtain fresh materials. Preliminary diagnosis suggested that our institution’s *A. mellifera* colonies had a lower SBV prevalence than our *A. cerana* colonies. This trial was designed to confirm that injections of the cloned SBV-IRES-EGFP RNA can infect the bees, and to obtain fresh materials for Western blot analysis and ribonuclease treatments. In addition, we added a negative control in the trial to verify the reliability of the clone and provide a better control group, bees injected non-infectious viral RNA. We created a single nucleotide mutation at the 4008 bp of the cloned SBV genome as a mutated stop codon.

Two clones were inoculated, the negative control and the SBV-IRES-EGFP clone. The cloned viral RNAs were transcribed and injected into *A. mellifera* pupae. We used pupae instead of larvae because the pupae appeared to have less virus infections in preliminary diagnosis and for ease of manipulation. The amount of the injected RNA was increased to 2000 ng per pupa because of the size difference and a previous report (3) suggesting that higher dosages of the Ac-SBV strain might be needed to infect *A. mellifera*. In each group, 14 pupae, white eye stage (approximately 1-day-old pupa), were injected. Three groups were included in the trial: a control with no viral RNA, a negative control, and an experimental group. The pupae were held in a 34 °C incubator for 5 days before examination. We did not try to passage the clone in *A. mellifera* because all colonies were infected by DWV or BQCV, making it difficult to prevent contamination.

### 2.5. RT-qPCR for Estimating Virus Quantities

We designed two specific primer sets for SBV and IRES (Appendix A) and used the SYBR green method in the two-step absolute RT-qPCR quantification and transcribed viral RNA as quantitative standards. The transcribed viral RNA was quantified in Nanodrop (Thermo-Fisher, Delaware, DE, USA), and the copy number was calculated. A serial dilution of the viral RNA samples was applied in the same reverse transcription.

Honey bee larvae have a high ratio of proteins and lipids in soft tissues that disintegrate in RNA storage buffer. This created some challenges for RNA extraction using the traditional Trizol method. Therefore, for the final oral inoculation of *A. cerana* (P2) and the RNA-injected *A. mellifera*, we used the Kingfisher (Thermo-Fisher, Delaware, USA) magnetic processor with Labserv RNA kit (Thermo-Fisher, Delaware, DE, USA) RNA extraction method, but some larval samples were lost. Only one P0 *A. cerana* larva, used for inoculum, and two P1 larvae were included in the qPCR. CFX 384 touch (Bio-rad, California, CA, USA) was used for the qPCR, and all reactions were performed using ChamQ universal SYBR green premix (Vazyme Biotech, Nanjing, China) and 200 pM of each primer. The reaction conditions were 95 °C for 3 min, followed by 45 cycles of 95 °C for 10 s, and 59.5 °C for 30 s. The PCR efficiency of the SBV primer set was 101%, and the IRES primer set was 96.1%. The melting curve measurement was added after the reaction using preset conditions. We used the IRES primers resulting in a relatively large amplicon in qPCR because the primers partially annealed to the cloned SBV genome (forward primer) and EGFP gene (reverse primer), which may increase the reliability of the added sequence detection.

### 2.6. Western Blot of EGFP Expression

Protein samples were isolated from the abdomens of selected pupae. The dissected abdomens were homogenized in the buffer (8M urea, 1% SDS, and proteinase inhibitor cocktail) and centrifuged. Proteins in the supernatant were quantified using a BCA quantification kit. 60 µg proteins of each sample were boiled 10 min in loading buffer followed by SDS-PAGE using conventional methods and the Bio-rad mini-protean system. The proteins were transferred onto a PVDF membrane using the Bio-rad transblot semi-dry system. EGFP expression was labeled by mouse mono-cloned anti-EGFP antibody (Abcam, ab184, 1/1000 in concentration) and HRP goat anti-mouse IgG secondary antibody (Abcam, ab8224, 1/5000 in concentration). Beta-actin expression was used as the control for the protein samples, and labeled using anti-beta-actin with HRP (Abcam, UK).

### 2.7. EGFP Mutation Screening Using Sanger Sequencing

EGFPF (9174F) and EGFPR (9698R) were designed to amplify a 524-bp region of the added EGFP gene (725 bp) of the clone. We used Phanta Max (Vazyme, Biotech, Nanjing, China) DNA polymerase in the reaction condition: 95 °C for 3 min, followed by 45 cycles of 95 °C for 10 s, 59 °C for 30 s, and 72 °C for 45 s. Not all cDNA samples of the inoculated larvae yielded a single, ample (>50 ng/μL) PCR product that can be sequenced directly with the PCR primers. We obtained the sequencing results of six larval samples from the last oral inoculations. The EGFP sequences from the transcribed viral RNA that was used for larval injection was sequenced using the identical method for comparison. The sequencing chromatogram was compared using Poly Peak Parser [21] to identify ambiguous peaks.

### 2.8. Ribonuclease Treatment of the Cell Lysates from Infected A. mellifera Pupae

Homogenates of five clone-infected *A. mellifera* pupae were pooled and mixed with 5 mL RNaseA buffer (0.1 mg/mL RnaseA in 25mM Tris-HCl). The mixture was incubated at room temperature for 30 min and then centrifuged at 13,000× *g* for 10 min. The supernatant was collected and centrifuged at 141,000× *g* for 2.5 h. The precipitate was subjected to RT-qPCR amplification and analysis.

## 3. Results

### 3.1. Viral RNA Injection and Oral Inoculation in Apis Cerana Larvae

Of five larvae injected with the cloned viral RNA (P0), three survived and were evaluated using SBV and IRES-EGFP specific primers in RT PCR. The trial was terminated at three days post-inoculation (dpi) before the injected larvae entered the prepupal stage. No typical SBV infection symptoms were noted, but EGFP fluorescence was observed under a fluorescence microscope. No EGFP fluorescence or SBV was found in the control group (*n* = 20), which were individually examined using RT-PCR.

To enrich the clone and determine if it was infective by oral transmission, one positive P0 larva was selected and served as the inoculum for the P1 passage. Of 20 larvae surviving oral inoculation, 16 had SBV and EGFP sequences detectable in RT PCR screening. No SBV was detected in the 20 surviving control larvae. Body sizes varied in the inoculated larvae but not in the control larvae when the trial was terminated at 4 dpi. EGFP fluorescence was observed in the inoculated larvae under a fluorescence microscope.

Larvae in the P2 passage (inoculated from one P1 larva) were observably smaller than the controls (Figure 2); however, larvae were not weighed or measured. Eighteen of the 48 inoculated larvae were moribund or dead 2–3 dpi; 32 of the 36 control larvae were alive at 3 dpi. The trial was terminated at 3 dpi to avoid further mortality. Some dead larvae showed symptoms similar to typical SBV symptoms observed in the field (Appendix A). Fourteen inoculated and 12 control larvae were then fixed in formaldehyde, while the rest were homogenized for RT-PCR exams. All inoculated larvae were positive for the clone in RT-PCR tests or fluorescence microscope exams (Figure 3A,B). Larvae from the control group were all negative in RT-PCR or fluorescence microscope exams (Figure 3C). The produced virions were identified under TEM using a negative staining method (Appendix A). A third passage (P3) was attempted but the control group was infected by SBV without EGFP expression. We evaluated the *A. cerana* colonies again and found all colonies had detectable SBV.

### 3.2. RT-qPCR Exams of A. cerana Larva Infections

The virus-copy estimate of larval cDNA (1 μL) generated from viral RNA injection was 7.38 × 10^7^. Assuming that there was no loss in RNA extraction and reverse-transcription processes, the virus quantity of the whole larva can be calculated using the dilution factors and volumes (approximately 200× in total): 1.48 × 10^10^ for the whole larva, which is smaller than the copy number of the injected viral RNA (700 ng, 1.34 × 10^11^ copies). Although the naked viral RNA was probably digested in the larvae during the incubation period, we cannot exclude the possibility that the injected viral RNA contaminated the larval RNA samples. For the oral inoculation passages, 1/200 of the larva homogenate was used as the inoculum, which corresponds approximately to 7.38 × 10^7^ copies of the clone. In the larva with the highest RT-qPCR values in the P1 group, the virus quantity was 3.22 × 10^10^ (whole larvae, estimated). For the P2 group, 1/200 of the homogenate from this heavily infected P1 larva was used as inoculum. Two larvae of the P2 inoculation contained more virus copies (3.46 and 4.66 × 10^10^) than the inoculum, while the rest had fewer virus copies (Table 1). Overall, the clone proliferated in the orally inoculated larvae.

To determine if the added IRES-EGFP remained in the cloned virus through successive passages in *A. cerana*, we conducted RT-qPCR of the IRES fragments in inoculums and P2 larvae. IRES primers amplified the inserted BQCV IGR-IRES and partially annealed to the 3’NTR and EGFP. The RT-qPCR readings were lower than that of SBV primers but quite similar in absolute quantification when transcribed viral RNA were used as standards. RT-qPCR results for inoculums and P2 group are listed in Table 1. Dividing the qPCR results for the SBV by the results of IRES yielded a ratio for the third inoculation of 1.18 ± 0.44 (Table 1), not significantly different to 1:1 (*p* = 0.69, *t*-test), which suggested that the SBV genome and the added sequences remain in a 1:1 ratio in the clone. The added IRES-EGFP was not deleted through the passages.

### 3.3. Viral RNA Injection in Apis Mellifera Pupae

The inoculation trial from viral RNA to the first generation virus was repeated using *A. mellifera* pupae. We chose *A. mellifera* because SBV-free *A. cerana* were unavailable for more than a year (unpublished survey data). *A. mellifera* is known to be less susceptible to sacbrood disease than *A. cerana*. In our pre-trial evaluation of *A. mellifera* colonies, we found that 5–20% of the tested bees had detectable SBV, but all yielded low copy numbers in qPCR. In comparison, 60–100% of tested *A. cerana* had detectable SBV. We added a negative control to the *A. mellifera* studies, a clone with an additional point mutation resulting in a stop codon at the 4 kb region of the cloned SBV genome.

Inoculations of *A. mellifera* pupae yielded similar results to those obtained in the P0 trial of *A. cerana* larvae. In the inoculated group, 5 of 14 pupae showed positive results for both SBV and the added IGR-IRES EGFP sequences (1 μL cDNA had 2.36 × 10^2^ to 7.74 × 10^3^ copies) and two cDNA samples had SBV in low copy numbers (7.2 and 36). The control group showed no positive results. One of 14 in the negative control group exhibited positive but low copy results for SBV. We sequenced a short fragment from these infections using PCR products (3400F and 4420R, listed in Appendix A). The sequences were identical to our clone, but two SNP were found in the low copy infections. Because these two SNPs were not noted in the clone infected *A. cerana* larvae, we considered the low copy infections in *A. mellifera* to be field contamination.

### 3.4. Western Blot Analysis of EGFP Expression in A. mellifera

Three clone-infected pupae and three randomly selected uninfected pupae of the control and the negative control groups were used for the Western blot analysis of EGFP expression. Figure 4 shows that all infected pupae from the experimental group exhibited strong EGFP expression, while no EGFP expression was found in the control and the negative control groups. In addition to verification of EGFP expression in the infected pupae, the negative controls did not result in detectable EGFP expression in the host under the same conditions.

Three *A. cerana* larvae in the P2 group were also analyzed using Western blot. Because these samples were stored in a freezer for more than a year and experienced successive freezing and thawing, the detected EGFP expression (Appendix A) was not as strong as for *A. mellifera*.

### 3.5. RT-qPCR of the Ribonuclease Treated Cell Lysates from Infected A. mellifera Pupae

RT-qPCR using the IRES primer set found approximately 2.8 × 10^3^ copies in 1 μL cDNA in precipitate from the homogenate of pupal abdomen after RnaseA digestion. This is similar to that found in the cDNA generated from homogenates without RnaseA treatment. Although there were some differences in the dilution factors between the homogenates and ribonuclease treated lysates, the similar results suggested that IRES sequences are retained in the virions and protected from the RnaseA digestion.

## 4. Discussion

We established a novel procedure for studies on virus transmission and pathogenesis in honey bees. We were able to establish an infectious *A. cerana* SBV virus clone coupled with EGFP expression, which allows visualizing infection and the presence of the cloned virus in honey bees. The cloned viral RNA successfully infected *A. cerana* and *A. mellifera*, and the reproduced clone virus was passaged in *A. cerana* larvae by oral inoculation. The symptoms were replicated, which fulfilled the requirements of Koch’s postulates. The differences between *A. cerana* and *A. mellifera* also fit the expectations of the Ac-SBV strain, suggesting that the clone has similar biological characteristics to that of wild-type SBV.

The lack of clones or pure virus isolates is considered to be the major factor limiting research on honey bee virus research [9,11], but we noted that the lack of suitable hosts was also a limiting factor. Although a previous survey suggested that up to 90% of *A. cerana* colonies carry SBV [19], we were fortunate to have a few *A. cerana* colonies with no detectable common viruses when we initiated this study. These SBV-free *A. cerana* colonies, however, were no longer available after 2018, thus hindering part of our study. After inoculating the clone orally to larvae from these virus-infected *A. cerana* colonies, wild-type SBV appeared to dominate in the co-infection competition with the clone, but we were unable to analyze the results in detail.

We completed the final trials of the study using *A. mellifera* as an alternative host with a lower prevalence of wild-type SBV. The pupal stage was selected for the experiments based on relatively low virus titers and ease of manipulation. Naked viral RNA was injected directly because honey bee cells can intake exogenous RNA fragments [22]. The clone successfully infected *A. mellifera* pupae and expressed EGFP. EGFP in the negative control was not detected in the Western blot, which suggested that detectable EGFP expression may be only found in infective clone. Although the negative control has intact IRES-EGFP, the second ORF, which can express EGFP [23], EGFP expression was not detectable after 5-day post RNA injection.

The RT-qPCR results suggested that the IRES-EGFP is maintained in the clone. RT-qPCR using IRES primers yielded results in the P2 group comparable with those obtained using SBV primers. These results indicate that the added IRES-EGFP was not easily excluded during proliferation and oral transmission. In each orally inoculated larva, the cloned virus reproduced more than a few times to spread through tissues within the host; therefore, the number of recombination exceeded the number of passages. Because the added EGFP gene is not necessary for the survival of the clone, we were expecting many mutations to be detected in Sanger sequencing chromatogram data. However, no mutations were detected in the added EGFP gene. We also analyzed the SBV genome fragment (3400–4420 bp) in P2 infected larvae and no mutations or SNPs were noted. The SBV clone did not appear to quickly accumulate mutations or delete unnecessary sequences during replications within the host, but further investigations are needed to demonstrate the stability of the gene.

Although the present experiments were not designed to investigate the infectivity of the SBV clone in *A. cerana* larvae, some results provided cues about infectivity. The highest copy numbers found in the oral inoculation passages conducted to enrich the clone were 3.2–4.6 × 10^10^ in one larva, which might be near the number of plateau phase infection. The much higher dosage used in the P2 oral inoculation did not yield higher detectable virus numbers in all larvae. In addition, the dosages used in the trials suggested that the infectious dosage of our clone needed to reach ID_100_ probably exceeds 1 × 10^8^ viruses. Although the ID_100_ seems quite high for a virus-causing serious disease, it was similar to that of a wild-type DWV [24].

The bicistronic design is a simple method for adding a reporter gene in honey bee viruses. Published honey bee virus clones have mutations that were used as labels detectable only in RT-PCR analyses [10,11,12]. In this study, we inserted 908 bp after the 3′UTR, and the clone was still infectious to host larvae and stable through passages. The added sequence with IGR-IRES and an ORF resulted in a dicistrovirus-like genome structure: two ORFs separated by an IGR-IRES originating from BQCV. IGR-IRES had a similar expression effect as the IRES in the 5′-UTR [23].

SBV 3′-UTR and the added IGR-IRES EGFP apparently did not affect the replication process of the clone. 3′-UTR may have an essential role in RNA-dependent RNA replication of some picornaviruses [8], although the functions of 3′-UTR in replications are not yet fully understood [15,17]. The SBV 3′-UTR in our clone is relatively small, 82 bp, and simple in secondary structure predictions. Adding IGR-IRES as an adjunct to SBV 3′-UTR may have created additional complexity in the RNA secondary structures. Surprisingly, our results showed that the added sequences are stably included in the clone replications. These results suggest that SBV 3′-UTR is manipulatable and not necessary as the initial sequence of the synthesized negative-strand during replication or the added IGR-IRES replaced the functions. However, this study was not designed to investigate the functions of SBV 3′-UTR and picornavirus replications; these are interesting results reported here through the trial-and-error process of constructing a viable clone with a reporter gene that was successfully expressed.

The constructed clone has a full SBV genome with no mutations; thus, it should have full capacity to generate the disease. The reporter gene did not fuse or link with any protein of the cloned SBV, and can be easily swapped for other reporter genes or other genes of interest. Overall, these features are advantageous and facilitate the creation of SBV or iflavirus clones with the desired reporter gene.

Honey bee hives can only thrive in a natural environment where they are exposed to unpredictable contact with viruses. In addition to honey bees, wild bees sharing the same environment may also contract and spread the viruses [25,26]. Research on viral transmission mechanisms among hosts that naturally have multiple virus infections is technically complex [9]. In this experimental context, an easy-to-distinguish infectious clone that reproduces in laboratory conditions constitutes an ideal tool for such studies.

## Figures and Tables

**Figure 1 viruses-12-00224-f001:**
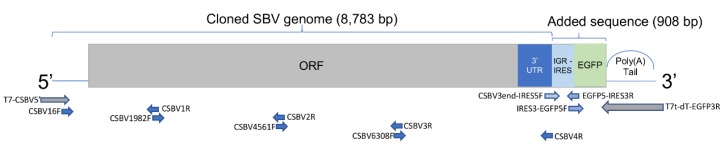
Diagram of the bicistronic SBV clone cDNA template. The bicistronic design was comprised of two ORFs, the ORF of SBV and IRES-EGFP as the second ORF. Black queen cell virus intergenic IRES (IGR-IRES) was used to induce the expression of EGFP. The cDNA template included a full SBV genome, and the untranslated regions (UTRs) of both ends were unmodified. Arrows beneath the diagram indicate the primers used to produce the fragments before assembling (dark blue indicated primers for SBV genome; light blue indicated added sequence; grey indicated terminals with modifications). The primers are listed in Appendix A. The full sequence, including partial vector sequences, has been submitted to GenBank (#MN528599).

**Figure 2 viruses-12-00224-f002:**
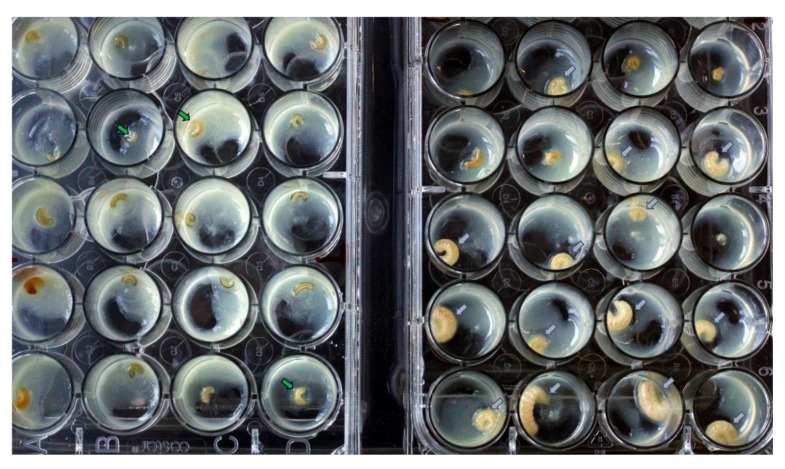
Observable differences between inoculated and control *Apis cerana* larvae in the second oral inoculation. The plate on the left shows the inoculated larvae; the plate on the right is the control group. Arrows indicate living larvae at 3 dpi. Of the inoculated larvae, 37.5% were dead at 3 dpi and the inoculated living larvae were smaller in size than the larvae in the control group The size difference shown in this figure is not described as an SBV symptom, which can be difficult to observe in a hive with different ages of larvae. Dead larvae showed SBV-like symptoms, color changes, and unshed cuticle; enlarged photographs are shown in Appendix A.

**Figure 3 viruses-12-00224-f003:**
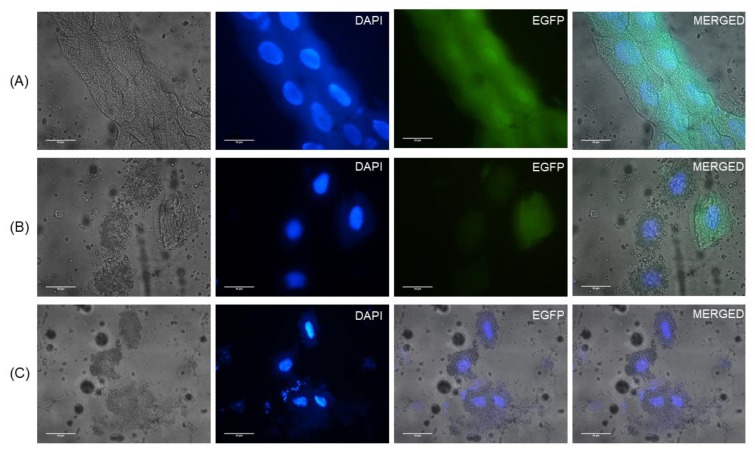
EGFP (green) fluorescence observation of SBV clone inoculated and control *Apis cerana larvae*. (**A**) Epithelial cells from an inoculated larva; (**B**) Oenocytes from an inoculated larva; (**C**) Oenocytes of a control larva. DAPI (blue) used as a counterstain. All photographs were taken under the same exposure conditions. Scale bars are 50 μm.

**Figure 4 viruses-12-00224-f004:**
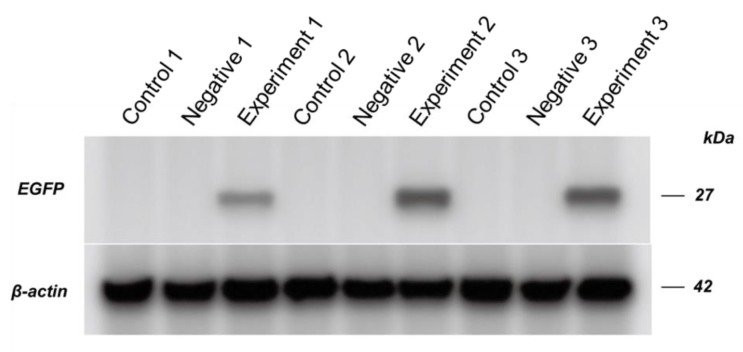
Western blot analysis of EGFP expression in clone viral RNA injected *A. mellifera* pupae. The control group pupae were injected with PBS; the negative control injected with a mutated clone with a stop codon; the experimental group was injected with the clone, SBV-IRES-EGFP.

**Table 1 viruses-12-00224-t001:** Absolute qPCR results. The quantifications were conducted using transcribed cloned viral RNA as standards. Two primer sets, one specifically designed for the cloned SBV genome and the other designed for the added IRES fragment, were used for the same cDNA sample, and the results were divided to show if the added fragment remained in the 1:1 ratio with the SBV genome. The numbers in the table indicate the virus quantities in 1 μL cDNA used for RT-qPCR; only the mean numbers of repeats were shown here.

Generation	SBV	IRES	SBV/IRES *
P0	7.38 × 10^7^	1.66 × 10^8^	0.445
P1	1.61 × 10^8^	1.01 × 10^8^	1.59
	3.84 × 10^7^	7.66 × 10^7^	0.501
P2	1.73 × 10^8^	6.80 × 10^8^	0.254
	2.33 × 10^8^	4.46 × 10^8^	0.522
	1.16 × 10^7^	2.67 × 10^6^	4.34
	1.58 × 10^7^	1.76 × 10^7^	0.898
	6.69 × 10^7^	4.94 × 10^7^	0.135
	1.41 × 10^7^	9.48 × 10^6^	1.49
	6.10 × 10^6^	4.68 × 10^7^	0.13
	1.03 × 10^7^	6.18 × 10^6^	1.67
	3.20 × 10^6^	2.67 × 10^6^	1.2
			P2 average: 1.18

* SBV quantification result divided by IRES result, showing the ratio of the two cDNA fragments.

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
