# Peer review of "Visualizing Sacbrood Virus of Honey Bees via Transformation and Coupling with Enhanced Green Fluorescent Protein"

_viruses, 2020, doi:10.3390/v12020224_

Round 1
Reviewer 1 Report
The revised version of the manuscript submitted by Long Jin et al., has improved than the first time I reviewed. I think, It is a valuable peace of new science in honey bee filed that can help to understand virus localization and transmission in future.
Author Response
1. The revised version of the manuscript submitted by Long Jin et al., has improved than the first time I reviewed. I think, It is a valuable peace of new science in honey bee filed that can help to understand virus localization and transmission in future.
I appreciate the comment on the behavior of all co-authors.
Reviewer 2 Report
The revised version of the manuscript "Visualizing Sacbrood Virus of Honey Bees via Transformation and Coupling with Enhanced Green Fluorescent Protein" by Jin et al. has significantly improved. After the first round of review the authors received quite some criticism, which they have taken seriously. Large portions of the manuscript were re-written and or improved. It became much clearer now what the main intention of the manuscript is. The methods are easier to follow and hence also the presentation of the results makes much more sense now.
I have some minor suggestions to make.
Table 1 contains as a third column of data the ration of SBV and IRES. I think you could use basic numbers (e.g. 1.1) instead of scientific terms (1.1E+00). That is simply confusing and distracting from the main message. What I don't understand is why you not deliver a statistical test, instead of having just the descriptive mean. You could simply use a single sample t-test to test against 1, which will probably not result in a deviation.
Line 397. there is a yellow mark, should be removed
Author Response
1. The revised version of the manuscript "Visualizing Sacbrood Virus of Honey Bees via Transformation and Coupling with Enhanced Green Fluorescent Protein" by Jin et al. has significantly improved. After the first round of review the authors received quite some criticism, which they have taken seriously. Large portions of the manuscript were re-written and or improved. It became much clearer now what the main intention of the manuscript is. The methods are easier to follow and hence also the presentation of the results makes much more sense now.
2. Table 1 contains as a third column of data the ration of SBV and IRES. I think you could use basic numbers (e.g. 1.1) instead of scientific terms (1.1E+00). That is simply confusing and distracting from the main message. What I don't understand is why you not deliver a statistical test, instead of having just the descriptive mean. You could simply use a single sample t-test to test against 1, which will probably not result in a deviation.
3. Line 397. there is a yellow mark, should be removed
Really thanks for this comment. I am really glad that the efforts of rewriting have paid off and hoping the future readers can understand the science and the results within the paper easily after it is published. The numbers were changed and the statics was added accordingly. I do not have strong background in statistics, and it was not the first solution appeared in my mind. Thanks for the suggestions. Corrected.
This manuscript is a resubmission of an earlier submission. The following is a list of the peer review reports and author responses from that submission.
Round 1
Reviewer 1 Report
The manuscript submitted by Lang Jin et al, used genetic engineering techniques and added two extra IRES-EGFP into the SBV genome and cloned the virus genome. It is a need in honey bee virology field to study viruses, where authors has done a nice job to integrate those two parts, but the bio-assay part could be designed more carefully than what is stated in this manuscript. Probably, repeating the experiment with more care can improve the results.
In general, the text is written in a fluent way, however the text need many grammatical corrections throughout the text to elevate the readability.
Introduction needs to be polished with more focus on the subject. This study by itself is important, and introduction need to tell a nice story and background about SBV. There are several important studies about SBV that has not been mentioned including (Procházková et al 2018 virion structure and genome delivery mechanism). There have been three clone of honey bee viruses (BQCV, DWV and CBPV) so far. It will be useful to discuss the advantage and disadvantages of the method in this study with previous studies. In addition the second half of the introduction is part of obtained results. I think, the authors need to write a strong background about their work than writing the results in the introduction part.
For a consistency manner with other publications "nontranslated region (NTR)" need to be change to "Untranslated region (UTR)".
Line 86: "after more than 24 h" is a bit vague. Please rewrite the sentence.
Line 87:RNA was extracted instead of RNA was obtained.
Line 116: the provided id number is not directing to the submitted sequence. Please be sure that the id number is correct.
Line 119: this section regarding the natural virus infection level is a bit confusing. First you mentioned that you chose three colonies but by the end of section it was stated that no virus free AC colonies were found after 2018. did this experiment perform within two different years? later in the result you stated that you could not find SBV free colonies in 2019. It is a bit confusing by stating after 2018 in material and method section and 2019 in the result section.
line 157: "In addition, ...." what was the reason?
Line 247: did you check other viruses from homogenized larvae before inoculating second round of larvae?
Line 226-227: are symptoms similar as the ones due to wild type of virus? Since you had access to AcSBV wild type it would be a good idea to not only negative control in your experiments but also positive control using virus wild type.
Line 304: you stated:"....for studies on pathogenesis and virus transmission". It is a good idea if you can explan your experiment base on Koch’s postulates
Reviewer 2 Report
This manuscript explores the development of a GFP-expressing SBV. The difficulty in this manuscript is the lack of replication with very little quantitation for the different experiments. Lack of data collection due to “unawareness” was mentioned. While the commentary provided why different decisions were made, it does not change the lack of data in many sections of the results.
Below are several further points that should be addressed.
Lines 54-55 - This sentence is confusing and needs editing.
Lines 135-138 - This sentence is confusing and needs editing.
Lines 143-144 – What year did the study begin?
Lines 157-159 – Why was it decided to not to produce the virus in A. mellifera? What does IC50 stand for?
Lines 198-199 – It is stated that not all samples yielded a single, ample PCR product. Does that mean there were multiple bands or that not enough cDNA was produced for sequencing?
Line 206 – homogenates not homogenizes
Line 217 – Sentence is unclear, “control group integrated by 20 larvae.”
Lines 267-280, 296-302 – Where is the data to support the work in this section?
Lines 346-356 – The bp size of the different portions of the cloned virus would have been helpful near Fig. 1.
Table 1 – What is the significance of the underlined/bolded values in the table?
Fig.1 – Add the SBV genome and expression tag size to the figure.
Fig. 2 – Legend states that dead larvae were removed but there appear to be dead larvae on both sides. The arrows (which are difficult to see) indicate living larvae? Arrows point to what looks dead in both the inoculated and control larvae.
Fig. S2 – Difficult to visualize produced virions in image. This doesn’t add any information to the manuscript.
Reviewer 3 Report
The manuscript “Visualizing Sacbrood Virus of Honey Bees via Transformation and Coupling with Enhanced Green Fluorescent Protein“ by Jin et al. deals with a methodological study showing that a picornavirus of honeybees, the sacbrood virus (SBV), can be manipulated to visualize distribution and transmission of the virus.
For me it was difficult to figure out whether this is a methodology paper or not. This should be stated more clearly. Furthermore, if it is a methodology paper, I would have awaited some success stories of the new methodology. I was surprised to see pictures of GFP under microscope in the supplemental material, but not in the main part of the manuscript. I think there is a very clear way needed that describes what you want to do. Is it just the description of a study to invent a new methodology, than you should aim for a different journal. Here, you would need much more way-forward-thinking. It is not understandable how you can evaluate your method, especially with the aim of visualization of a virus.
Furthermore, I found it quite confusing to differentiate between the original host, A. cerana, and the - probably - secondary host, A. mellifera. It is easily understandable that there is a problem with experiments when the original host is not any longer available ion uninfected status. However, just switching to another host species might not be the best option, as we don’t know the reaction (growth/replication rate, virulence, etc.) of the virus in the second host.
Overall, the experimental procedure is hard to understand, especially in face of the switch of host species and their utility to serve as alternative hosts. Furthermore, the visualization of the virus in the host seems to be the major aim resulting from the title of the manuscript, while the content focuses mainly on some methodological advances. Hence, it is difficult to judge about the rea advances of this study.
I suggest to re-write the whole manuscript and outline clearly the exact background ad aim of this study and finally outline the aims of this study.
Minor comments:
L 151: fresh instead of flesh
L 162: …34 C incubator… should be …34°C incubator…
L172: A. cerana and A. mellifera should be in italics
L 247: the larva RNA samples … change to …the larval RNA samples…
L269: 2019 . change to 2019.